# Exploring the Meanings of Food Sustainability: An Interpretive Phenomenological Analysis

Claudia Troncoso-Pantoja [1] , Paola Cáceres-Rodríguez [2,*] , Antonio Amaya-Placencia [3] ,
Claudia Lataste-Quintana [4] and Rodrigo Valenzuela [2,*]

1. Centro de Investigación en Biodiversidad y Ambientes Sustentables (CIBAS-UCSC), Departamento de Salud Pública, Facultad de Medicina, Universidad Católica de la Santísima Concepción, Concepción 4090541, Chile; ctroncosop@ucsc.cl
2. Department of Nutrition, Faculty of Medicine, University of Chile, Santiago 8370451, Chile
3. Facultad de Medicina, Universidad Católica de la Santísima Concepción, Concepción 4090541, Chile; aamaya@ucsc.cl
4. School of Nutrition and Dietetics, Faculty of Medicine, University of Chile, Santiago 8370451, Chile; claudia.lataste@uchile.cl
* Correspondence: paolacaceres@uchile.cl (P.C.-R.); rvalenzuelab@uchile.cl (R.V.)

**Abstract:** Sustainable food combines nutritional, environmental, economic, and social aspects. Considering the need to recognize the Chilean reality in this matter, this study proposes to interpret the meanings that food professionals give to food sustainability, based on the question: What meanings do food and nutrition professionals give to the relevance and measurement of food sustainability in their professional work? To answer this question, qualitative research was carried out using the interpretative paradigm of exploratory scope. Semi-structured interviews were conducted, and their responses were analyzed using the content-analysis technique. The results indicate that environmental care is valued in accordance with the local regulations in force. Despite this valuation, its implementation has been complex in developing countries, due to economic aspects and political will. Sustainable culinary preparation is identified, building a definition based on ecology and local identity, highlighting the use of natural foods. In addition, there is a need to standardize the indicators for its measurement and to reinforce communication strategies to improve its knowledge. In conclusion, for professionals in the gastronomic and nutritional areas, the sustainability of food and culinary preparation requires a comprehensive, integrated vision of the different factors, depending on the user and the entities that provide food services.

**Keywords:** sustainable diet; ecology; food system; restaurants; carbon footprint; water use

## 1. Introduction

Sustainability can be understood as the dynamic balance between a population and the carrying capacity of its environment, which enables its potential expression without affecting the environment on which the population depends [1]. Etiologically, the concept of sustainability incorporates socio-ecological, economic, and public policy issues related to the environment, and industrial and agricultural production, among others [2,3], showing the ability to adapt to the geographical location or objectives set, crossing generations with the human being as part of the biosphere [4]. A sustainable diet is one of the foundations of food systems and food security, and combines nutritional aspects and global environmental, economic and social sustainability, and is guaranteed only when there are no unexpected impacts, such as epidemics, economic crises and extreme climate, which limit the stable supply of food [5–7]. It can be considered in itself as healthier since it uses more sustainable foods in its culinary preparation that favor environmental awareness and promote the circular economy in the territories [8]. However, it is necessary to understand that one of the major complications involved in sustainable food is caused by the explosive population

growth in the current century; a situation that must be compensated for in order to maintain environmental balance, with a culinary culture that reduces food waste, greenhouse gas emissions, deforestation of large territories and, in the same way, reduces poverty [9,10]. To achieve sustainable food, several technological or systemic innovations have been recognized, which are necessary for food systems and range from food production to land use, emission of harmful gases, as well as improvements in diets, modifications of dietary patterns, and waste management [11,12]. Recognition of the importance of sustainable food must be accompanied by a model or tools to measure long-term sustainability. The European Union assesses sustainable food and nutrition security through its SUSFANS program (metrics, models and foresight for European sustainable food and nutrition security) which is based on a model that measures the sustainability of food diets and food systems for the years 2010 to 2050, projecting and predicting food supply and demand and considering, among other aspects, the market or existing sustainability policies [13]. Previously, a tool measuring the ecological indicators of fruit and vegetable consumption (EIFVC) was built, with a methodological approach to select and measure the most relevant EIFVC at the local scale [14]. This tool was applied in France and provided information about the three phases of consumption: acquisition, use, and disposal of waste, and the upstream and downstream phases of the consumption process, considering also the local and household ecological impacts [14]. However, the measurement of sustainability indicators presents a methodological diversity that makes their comparison difficult for the classically measured variables: economic, environmental, and social [15]. In Latin America, research on food sustainability is incipient, with measurements of sustainability indicators in research conducted in Colombia [16], Ecuador [17] and Brazil, among other regions of the geographical area, although the methodologies used to measure their variables do not present unified criteria, a factor which maintains uncertainty at the time of measuring food sustainability. Faced with this scenario, in which the current literature identifies inequalities in the development and valuation of sustainable food and nutrition between developed and developing countries, showing gaps and different levels of progress between the different Latin American countries, the need arises for a tool to assess food sustainability, which presents a community identity. The construction of a tool with a local identity that facilitates the evaluation, and the development of sustainable diets should be based on the opinion of professionals from different areas. Therefore, this research proposes an advance in the interpretation of the meanings that food professionals give to food sustainability, referring to their understanding of sustainable diets and culinary preparations and their possible measurement, as well as indicators of sustainability.

## 2. Materials and Methods

This research focuses its interest on answering the following question: What meaning do professionals in the area of food and nutrition give to the relevance of food sustainability in their professional work? The study is based on the hypothesis that professionals in the area of food and nutrition positively value sustainable diets and culinary preparations, as well as recognizing the need for indicators to measure the sustainability of these diets and dishes.

To respond to this question, the research approach was qualitative from an interpretive paradigm of exploratory scope, which allows the construction of reality from a dialectical reality in a specific context, not widely studied in the country where it was carried out [18]. The objectives of the study deepened and respond to the understanding of sustainability in food services or collective catering: culinary preparations that are recognized as sustainable and the analysis of sustainability indicators, specifically their understanding and standardization, as well as the internal and external communications delivered to the community.

As part of the methodology, the sampling was non-probabilistic for convenience based on the selection criteria of the study, which considered professionals with experience of at least 5 years, in the area of food and/or a sustainable diet, in addition to signing the

informed consent, an instrument validated by the Ethics Committee for Research in Human Beings of the Faculty of Medicine of the University of Chile (105-2021), which guarantees the handling and support of the data obtained, as well as the confidentiality of the latter. The sample was obtained by theoretical saturation; that is, when new additional data is not found to develop properties of the category by replicating similar instances of categories repeatedly, it becomes empirically sure that a category is saturated [19]. In this investigation, the saturation was completed with 15 experts from Chile, Colombia, Brazil, Argentina, and Spain (Figure 1).

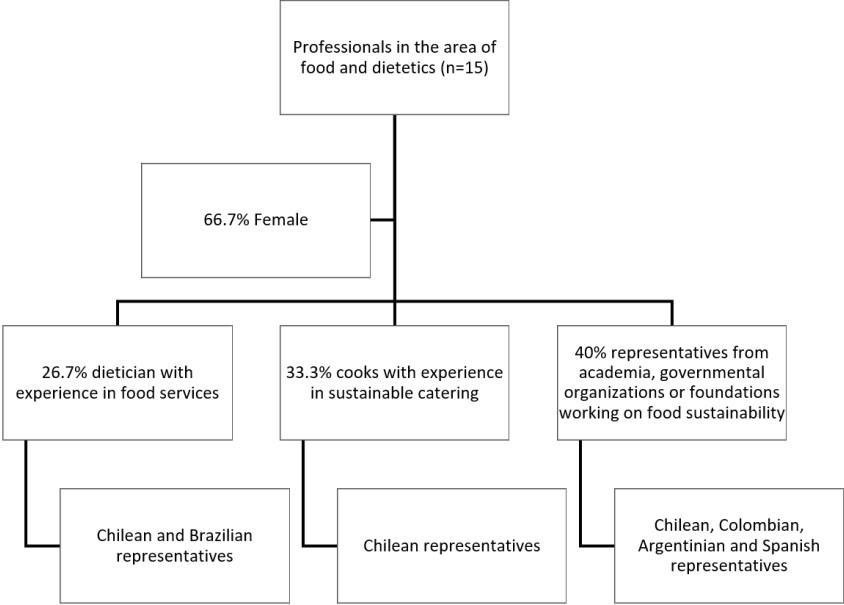

**Figure 1.** Participant description.

For data collection, a semi-structured interview was used; a flexible instrument that starts from a structured question, but that adapts to the participant's response [20]. The instrument was developed by the executing team, then subjected to validation by experts in the area and a pilot test before being applied. The items that make up the instrument are shown in Figure 2. The field work was carried out by a sociologist with experience in qualitative research, supported by a trained nutritionist, who recruited the participants. The online interviews were conducted through the Zoom© or Meet© platforms, recorded in video format and then fully transcribed.

> **Questions asked in interview**
>
> • What is your opinion on how sustainability is being addressed in collective food/catering services? Do you know of some sustainable venues or in the process of becoming so?
> • At present, could you identify if a collective food/catering service is sustainable? How?
> • What do you understand by sustainable culinary preparations? Give some examples based on your answer.
> • Do you know of some indicators or tools to identify a culinary preparation as sustainable?
> • Do you think it is necessary to have a standardization of indicators that qualify a culinary preparation as sustainable?
> • If such a standardization or tool existed, what attributes should it have so that its implementation and use contribute to the sustainability of food systems?
> • What is your opinion of the communication that must be made with diners/users of food services (casinos, restaurants) regarding the sustainability of the culinary preparations offered?
> • Do you think that communicating the level of sustainability of culinary preparations to diners would have a positive effect in promoting their consumption? Because?
> • What benefits do you see from being able to qualify the preparations as sustainable?

**Figure 2.** Items of the instrument elaborated to guide the semi-structured interviews.

The research data analysis process was carried out using the content analysis technique, which organizes and extracts meanings from the collected data and, at the same time, draws realistic conclusions from these data [21]. This activity was carried out by a

sociologist with experience in qualitative studies and triangulated by a nutritionist, part of the executing team, also with experience in qualitative studies. The data management strategy began with an ordering of the information; that is, identifying the common themes that arose from the participants' discussions, which allowed the identification of the respective codes and categories. The information was rearranged to carry out axial coding, which allows the answers to be ordered according to category and subcategories so that later, a selective coding could be carried out, which allowed the development of the respective interpretations in order to obtain the results of the study.

Like all research, this study presents a plan that safeguards the quality of the data and its processes. For this, the quality was safeguarded through the criteria of consistency, validity, and confirmability. The data were consistent since they were obtained under a strict and orderly methodological protocol, which included a validated data collection instrument before being applied, as well as roles in the research team and its collaborators. The validity or credibility criterion were met through the data analysis protocol, limiting the impact of the implementing team's own beliefs and through the triangulation of data between researchers, which allowed the participants' thoughts to be contrasted. The confirmability criterion was met by detailing the selection of participants, conducting interviews in coordinated virtual spaces and presenting the authorization for their conduct to an ethics committee. In view of the above, the quality of the research is relevant, since it contributes to knowledge within its object of study, in a context that includes professionals from various professions and countries [22].

### 3. Results

#### 3.1. Sustainability in Food Service and Collective Catering

To understand the concept of a sustainable food establishment, it is necessary to identify, from the perception of the participants, if a culinary preparation or a sustainable diet represent a geographical region of a country or if it implies that a place bases its gastronomy on native foods; a situation that would not only favor biodiversity, but also support the small local food producer, thereby reducing transportation and the emission of greenhouse gases, and ensure a supply of seasonal foods that also guarantee adequate nutritional value.

According to those interviewed, the environmental aspect stands out as an essential part of sustainability in collective food and restaurant services. It is an element that must be understood from a paradigm that integrates the crucial factors for society, the economy and also the environment, since it has an impact on public policies in the medium and long term that impact the ecosystem and food, as well as food environments that are healthier for the population. However, the participants recognized that the approach to sustainability is still in progress and for which there are many gaps, with a lack of knowledge and a very limited vision at the community level and especially in companies in the gastronomic or food area. It was also mentioned that a connection is necessary between the regulations of the states and territories and the execution of and compliance with the legal regulations that allow the declaration of sustainability of their processes with companies from various sectors, including gastronomy. However, as a criticism of national and international organizations, a local lag in food production is amplified since the institutions favor or encourage the production of local or cultural food that, in the end, is not consumed by the corresponding population, increasing the lack of a sustainable food system.

Regarding the understanding or identification of a food or catering service as sustainable, most participants interviewed declared that its implementation is especially complex because of the various premises that must be resolved in order for this particular group to accomplish the intention of delivering a sustainable food or catering service rather than genuinely being a sustainable gastronomic business. From experience, and derived from the opinions of the participants, it was shown that there is a difference between what is declared as sustainable and what is actually implemented as a sustainable service at a local level. The fulfilment of various actions that must be addressed to recognize an

establishment as sustainable, an especially limited focus, is understood in the economic aspects of its foundation or in other factors like the individual commitment of teams that form a part of the aforementioned establishments. This experience, however, does not limit the acknowledgement of other realities in countries and cultures that have been able to implement sustainable gastronomic establishments.

In actuality and at a local level, the importance of having sustainable food or catering services, from a social standpoint and focused on rights in the same way that they should be integrative, innovative and democratic, has been identified but these principles are obscured when their implementation is seen as demanding, particularly in a public context and dependent on the type of establishment; in Chile, as of a few years ago, there was nationwide legislation in regard to corporate responsibility for food quality and waste systems but the legislative power has not yet selected laws that adequately protect the ecosystem.

### 3.2. Sustainable Culinary Preparations (Dishes)

Another relevant aspect in the understanding of sustainable diets was established from the perspective of experts in relation to the concept of sustainable culinary preparations. The definition of this concept is exemplified by a preparation that presents harmony with the ecosystem, local food systems and cultures, and contemplates a low environmental impact that includes a reduced water and carbon footprint. These preparations must contain natural products and consider that the sourcing of the supply chain, as well as the supply chain itself, is local, seasonal, and healthy. In relation to the ingredients of these sustainable culinary preparations, experts have stated that the foods used must be sustainable in themselves and preferably of natural origin, since this promotes the healthiness of the preparation as well as being of local production and identity which in turn reduces the carbon footprint.

However, stating these as sustainability principles raises the question of what is the objective of the idea of sustainability or more specifically, who is this conceptualization directed toward: the environment? the people? the economy? the cultural tradition of food? Considering this, sustainability, beyond just the preparation, is once again seen to be multifaceted which implicates various areas of society, transforming it into a more complex and difficult phenomenon to classify by a single evaluation. By way of example, a participant mentioned that the concept of sustainability is weakened by its definition since it is possible to understand that a kitchen is sustainable when the whole broccoli is used (florets and stem), does that alone classify it as sustainable? just because the stem is not thrown away and is given a second use? This opinion must be understood with caution since the sustainability of a kitchen is determined by a series of sustainability indicators as a whole and not by a single isolated indicator.

Along the same lines, it was reported by participants that the packaging materials used for the delivery or presentation of dishes are intertwined with the life trajectory presented by the consumer, based on the perception that younger consumers have an increased rejection of the use of plastic bags or Styrofoam, for example, due to their unsustainable nature.

A shared idea among the participants was the understanding that sustainable culinary preparations should be recognized by their local identity, inherited through generations, and seen as endemic so as to humanize the diet and at the same time, maintain biodiversity and support local producers. They emphasized the importance and impact that transportation has on the greenhouse effect and of a reduction in the production of food waste. This aspect also reflects that the local and endemic significance represents a different viewpoint of development, beginning with one's own idiosyncrasies, values, and culture which is why, for the recognition of a sustainable culinary preparations it is necessary to pay attention to the customs and history of each country from their subjectivities and experiences that allow a sustainable regional gastronomy.

### 3.3. Sustainability Indicators: Knowledge, Standardization, and Communication

Taking as a reference the debate presented in the mentioned cases, regardless of the nationality of the participants, it is possible to determine the key indicators of sustainability such as the water and carbon footprint, the use of water, the composition of the diet considering whole grains and types of meat, loss and waste of food, promotion of local or community production, dishes prepared in the domestic food environment and transport. The latter connects again with sustainability, in this case, the proximity of the place of origin of the food, highlighting the importance of local and territorial production as a source of sustainability.

The change in seasons created tension among the participants since one point of view claimed that the use of fruits and vegetables based on their seasonal availability for the planning of menus, for example, in a restaurant, does not occur, given the gastronomic demand to produce culinary dishes independent of produce seasonality.

From the claims presented by the participants, it was also stressed that there is a need to comprehend sustainability indicators as not just the raw materials used: non-food items comprising detergents, clothing, cleaning supplies, among other things, because of their necessity in a space that produces food, as well as the identification of the final destination of waste and surplus, the cost of energy, fair pricing, and the dimensions that surround the inherent processes of producing and consuming food.

With respect to the standardization of sustainability indicators, the participants completely concurred with the notion that there needs to exist a tool that recognizes collective food or catering services as sustainable; this identified a gap between understanding and methodology in relation to the instrument of information collection. This opinion was maintained regardless of the nationalities of the participants.

It has been suggested that a homogenous tool would favor competition between establishments that present quality evidence for their processes. In this sense, the attributes considered for the design of this tool, as mentioned by the participants, should be practical and easy to implement. For this, they suggested that there could be a trial run or even an adaptation of a national/international experience and that could be applied in all areas of production of a culinary dish in catering: the reception of raw materials, storage, refrigeration or freezing, temperature control, elaboration, conservation, distribution, and consumption. With the creation of a standardized tool, opinions would revolve around similar levels of information, resulting in education for the consumer and future professionals in the industry. However, some participants have raised concerns for this process, concerning the possibility of it generating unequal and complex competition simultaneously since every establishment would be measured against the same standards, from restaurants to collective catering services. It was suggested that the tool should be differentiated depending on the type of establishment for appropriate implementation to avoid this situation.

Sustainable gastronomic establishments should present a proper marketing and communications plan, according to the participants in our survey. They pointed out the necessity of including and involving all the components of the process that create a sustainable culinary preparation, in particular, the consumer who should know about the actions taken by the gastronomic business. Although this is primarily a challenge for the business, it is also a guarantee for increasingly informed consumers and for the ecosystem, additionally allowing for the promotion of education that goes beyond the simple dissemination of awareness.

Following on from the above, for the information about the sustainability of culinary preparations to achieve the desired positive effect on the consumer, the need arises for the communication to be effective, in addition to allowing significant lessons to be learned by the consumer. Communication is essential as it enables the modification of opinions and, in that way, acts upon the attitudes, resulting in changing the way people act. As a methodology of communication, it was suggested that gastronomic establishments give

their information to customers in statements on menus, on the interior of their physical locations, and on social media.

### 3.4. How to Measure the Sustainability of Culinary Preparations (Dishes)?

The use of a tool or an instrument for sustainability evaluation is recognized as an appropriate way to communicate to consumers and the general community about indicators that address sustainability, like the carbon footprint or purchase of locally produced ingredients, among others. The participants believe that this tool should be comprehensive and grant the ability to clearly and quickly identify a food establishment with sufficiently sustainable practices.

One of the options that garnered a positive view and opinion from the participants of the study was a rating that utilizes stars because of its popularity and the simplicity presented by this evaluation methodology, but some showed opposition to the style of this tool and its adaptability along with the possible confusion with hotel ratings. It was also mentioned that they should be easily understood by consumers, using ecological shapes or forms related to nature, mainly from the viewpoint of the chefs, such as leaves or plants or even the adaptation of the symbols to represent diverse locations. In any case, there must exist a clear understanding and recognition of the concept of sustainability and its value that leads to sociosanitary care and the protection of ecosystems. Figure 3 shows the main results obtained from the interviews conducted with the participants.

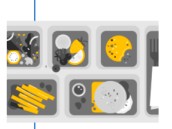

**Sustainability in food service and collective catering**
- It highlights the environmental aspect as an essential part of sustainability.
- It is recognized that the approach to sustainability is still a work in progress and that there is a local gap in food production; the production of local food, which in the end is not consumed by the population itself, is promoted and encouraged.
- There is a difference between what is declared as sustainable and what is finally implemented as a sustainable service at the local level.
- The implementation is difficult, at least in Chile, due to economic aspects or other factors such as the individual engagement of human teams.

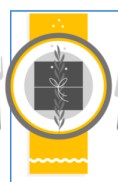

**Sustainable culinary preparations**
- It is interpreted as those that present a harmony with the ecosystem, local food systems and cultures, as well as a low environmental impact.
- They should preferably be composed of sustainable and natural ingredients, which enable them to be healthy and with local identity and acquisition.
- Emphasis is placed on local identity, intergenerational transfer, heritage and endemism, humanizing food while maintaining biodiversity and supporting small producers.
- It is recognized that its definition is complex and that it is difficult to classify it in a single valuation.

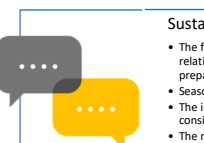

**Sustainability indicators: knowledge, standardization and communication**
- The following indicators are recognized: water and carbon footprint, water use, dietary composition in relation to whole grains and meat type, food loss and waste, promotion of local production, culinary preparations made in domestic food environments and transportation.
- Seasonality creates a space of tension in the participants.
- The inclusion of not only the food ingredients used is addressed: non-food products should also be considered.
- The need for a tool that recognizes collective food or catering establishments as sustainable appears; a lack of knowledge and methodology on some instrument is identified.
- There is a need for effective communication, for example, through information on menus, inside the establishment and social networks.

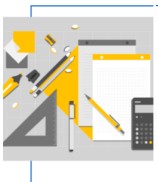

**How to measure the sustainability of culinary preparations?**
- It is recognized that the use of some kind of sustainability assessment tool or instrument would be appropriate to communicate to consumers and the community in general.
- A measurement tool should be comprehensive and allow to clearly and quickly identify a food establishment with adequate sustainable practices.
- The star rating appears, due to the popularization and simplicity it presents, although it is warned that it could present confusion with the valuation of hotels.
- For its implementation, there must be a clear and known understanding of the concept of sustainability and the valuation it entails.

**Figure 3.** Main findings in the discourse of the participants grouped by themes.

## 4. Discussion

Based on the opinions and subjectivity of the participants in the present investigation, it could be interpreted that the sustainability of diet and culinary preparations should be based on an integrated vision of the various factors that are dependent on the consumer and the entities that offer food services.

It is interesting to note the support and outlook generated by those in the areas of gastronomy and nutrition, principally, their focus on the processes and presentation of diverse culinary preparations that possess local identity and a global center, encouraging waste reduction; a situation that has previously been reported in other studies [23,24].

A fundamental aspect in food sustainability, which was recognized by the professionals participating in the study, focuses on the relevance given to food waste and disposable packaging. Research led by Topole et al. [25] features this situation, highlighting the importance that consumers at various local gastronomic celebrations assigned to non-sustainable disposable materials that were disposed of at these events. Although daily practices related to food generate waste [26], domestic food environments are the primary environments where food waste is generated [27], unlike restaurant establishments, that acknowledge and strive to reduce food waste [28]. This situation creates tension that should be considered as consumer information, since, in the collective imagination, there could exist a construct of visualizing companies in the gastronomic sector as the largest polluting agents in the ecosystem; a situation already mentioned as an assumption [29], although the use of plastics or other disposable materials in places that deliver takeaway meals should not be excluded, where an increase in waste originating from these containers is recognized [24,25]. This is a situation that the gastronomic industry has also visualized and largely solved with the use of materials of organic origin [30,31].

However, the relevance given to this by the study participants is not shared by the authors of other investigations. Among others, Fardet et al. [32] declare that global food systems are no longer sustainable for health, the ecosystem, biodiversity and animal welfare, culinary traditions, socioeconomics, or small farmers, due to the consumption of foods of animal origin and ultra-processed foods. On the other hand, Petrescu et al., conclude that the quality of food, mainly of plant origin, receives great attention from consumers and they relate it to health and care for the environment [33]. Both sets of authors agreed on the relevance of the quality of food or diets for sustainability and care for the environment, rather than on the assessment of waste made by the study participants. Initiatives or good intentions that are very local in the communities cannot be effective or have long-term effects if they are not accompanied by consensus among the interdisciplinary experts on food, health, and sustainability, which must be supported by public policies on the conceptualization of a healthy and sustainable diet [34].

The efforts made to achieve recognition for sustainable gastronomic establishments are for a goal that will be built over time so cannot yet present an answer, due to the diversity of the measures or approaches used in its evaluation, focused especially on ecological issues rather than on a holistic perspective [35,36]. This situation, recognized by the professionals participating in this study, could be resolved if the need for effective communication strategies is understood, which would allow establishments to make explicit, for example, the way in which waste is recycled or how the casino or the restaurant saves water or energy [28]. The dissemination of scientific communications originating from casinos or restaurants can be carried out in a simple and inexpensive way, for example, by using ICT and through applications and/or social networks, which are tools that are widely used in the gastronomic field and that link culture and food, thus improving the appreciation of food and sustainable diets [37–42].

Regarding the above, the participating professionals suggested the measurement of the water or carbon footprint for the evaluation of sustainability, a situation reported in the results of previous research [43,44]. The reduction in the carbon and water footprint is a global political, ecological, and economic necessity [45,46], so the recognition by the study participants is in tune with the needs of a sustainable gastronomic establishment.

However, the limitation of economic resources is an eternal and true barrier in developing countries for the real implementation of sustainable gastronomic establishments, so it is necessary to rethink innovative strategies of low economic cost for ways in which it can be delivered. This information, based on existing evidence, formal documentation, or scientific articles, should be recognized by the owners of gastronomic establishments and disseminated; for example, that the purchase of food produced or distributed in these premises reduces the carbon footprint [47]. Another strategy to publicize products, foods or dishes with a low carbon footprint might be the dissemination of the value of the consumption of home-cooked meals made with natural foods [48], the appropriate use of water in agriculture or changes in the type of crops grown [49], low or zero use of chemical agents in vegetables [50], among other actions that do not require great economic resources and that lead to an informed choice by consumers when deciding on a culinary dish or going to the establishment.

It is also necessary to highlight the value that the study participants assigned to sustainable culinary preparations, understanding this term as culinary dishes where natural foods are used, with cultural identity and where the producers in their communities have active participation. The association of the use of traditional or homemade meals with a more sustainable, healthy diet due to the use of foods in their natural state or minimally processed and with a lower ecological impact is evident [51–53].

Tourism or gastronomic tours, although they are essentially commercial enterprises, can be seen as an option to intertwine gastronomic culture with sustainable and healthy cooking, due to the preparation techniques and ingredients used, allowing the users to attend gastronomic establishments that contribute positively to the ecosystem and that deliver meals with cultural identity and in which local producers have intervened in the supply chain, thus strengthening the sustainable circular economy [54,55]. This situation was also recognized by the study participants, although previous research results recognized a certain resistance on the part of the consumers, who, in their food selection, do not value solely the sustainability of the food purchased [56], which ultimately increases the need to adequately communicate the relevance of the consumption of sustainable dishes or to seek new ways to disseminate their products, as, more recently, the use of artificial intelligence allows [57].

It is the task of specialists in gastronomy and dietetics, among other disciplines, to strengthen the connection between what is considered sustainable and what is understood as such in the collective imagination. The way in which it is communicated and the level of information that is received by the community are key to establishing this bridge of union in favor of human well-being and the ecosystem. One way to understand and systematize this information to make it understandable is precisely the approach to the different existing indicators, since they generate a frame of reference to be applied, if possible. In this sense, the systematization of this knowledge becomes more valuable if adequate standardization is achieved, based on existing knowledge, and the ways of capturing it in the current context, in addition to considering the value of the territory and the locality.

Although, globally, there are gaps in the understanding and identification of sustainable diets by the scientific population and the community in general [58], researchers agree on the need to circulate knowledge that allows for the proper selection of foods and culinary dishes that are beneficial to the community and the ecosystem, suggesting focusing attention on young people and on the potential health, social and environmental benefits that the selection and consumption of sustainable foods entail [59–61].

So, how to communicate the sustainability of culinary preparations? Study participants mentioned the benefit of using a tool that measures some indicators that respond to this concern. This opinion is in line with different instruments built for this purpose, such as the one proposed by Leach et al. [62] with a label for food and its environmental impact, which showed the carbon, water, and nitrogen footprints, measured with various calculation methodologies, in four types of labels: stars, traffic light, complementary nutritional information and in a detailed comparison label. The authors concluded that the

star label was the simplest and easiest to understand but less detailed in terms of traces, compared to the situation of the more detailed label, recommending this type of tool for responsible and informed consumption [62]. For the above, Clodoveo et al. [63] proposed for the Mediterranean Diet, a lifestyle recognized for being healthy and sustainable, the Mediterranean Index (Med Index), a tool corresponding to a front label, to be applied by the producers, which addresses nutritional, environmental and social aspects, pillars of sustainability and empowers the consumer to visually recognize them according to colors, recognizing blue as the most sustainable product [63].

Another relevant aspect of sustainable eating and diet is the social role that eating presents as such. This interweaving is essential for one person, a family, community, country, and territories alike. It directs behaviors and eating habits and responds to food environments, especially at home. The home is the primary place for the formation of eating patterns, where gender, religion, and cultural prohibitions all influence eating practices [64]. But we must be optimistic and visualize opportunities: we are participants in sociocultural change for sustainable and healthy diets, arising from a few food movements, which entail modifications to lifestyles that would return us to traditional diets, which, as we have seen before, are themselves sustainable [65].

Standardization, through the construction of a tool that enables the recognition of sustainable food establishments, offers an opportunity in favor of the internalization of knowledge with the corresponding objectification of and reduction in the different aspects previously addressed. In this sense, the collaborative integration of the different perspectives interpreted in this study seems essential to cover all the aspects that are valued for the recognition of sustainability measurement indicators. Developing countries have unique opportunities that originate from the past experiences of developed countries in this area; however, its implementation is fraught with problems due to the different realities and the real political interest in issues of food sustainability and environmental protection.

Can we visualize in Chile the use of a tool that brings us closer to identifying sustainable foods, preparations, or diets? The answer is difficult. As a country, we understand the importance of sustainability, but we have not yet been able to fully internalize its relevance within the community; it is at a point where personal, ecological, economic interests, among others, are intermingled. The results of this research are the first instances of the construction of a tool that would allow the client of a restaurant or a collective food establishment, to have more information for the choice of a healthier diet that is more respectful of the environment. It is a small step, but it is a good way to start.

*Limitations*

The limitations of this study include those inherent to the methodological design of qualitative research, especially the lack of extrapolation of its results and the small number of participants. In addition, being an exploratory study, it only defines the concepts, points of view and knowledge of the participants about the meaning given to the object of study.

Another limitation was that not all participants were Chilean, so that interviewees from Spain, Colombia, Argentina, or Brazil, could present a story that contributes to the object of study, but fails to represent the Chilean reality in the field of sustainable food. This limitation is complemented by the fact that 40.1% of the participants represented academic, governmental, or foundational experience working in the area of sustainable food. This could be understood as a more theoretical contribution than from work carried out in daily practice.

Another limitation is the lack of consideration that food sustainability is only one component of the overall assessment of the sustainability of food systems. It is necessary, in further studies, to go more deeply into the national and international regulations that allow access to food produced in an environmentally responsible way.

## 5. Conclusions

The knowledge and valuation of food sustainability and gastronomic preparations highlight a diverse evolution at a global level, which enhances and marks the inequalities between territories. These gaps harm communities and ecosystems, ultimately affecting the quality of human life and its environment globally.

We are all part of the ecosystem and its preservation, so organizations, policies, the media or training programs of educational institutions, among other actions, will be essential and form the basis for increasing awareness of our role in preserving food and health in the communities.

This research reveals the views held by professionals in the gastronomic and nutritional fields, which recognize that food sustainability and culinary preparations must take a comprehensive look at the various factors which are dependent both on the user and the entities that provide food services.

This interpretation poses socio-ecological and economic challenges for its implementation in a developing country. A comprehensive and integrated view of the processes and results must be generated, in connection with public policies that can accommodate these requirements and involve their actors and transmit this to the community and consumers, through an adequate communication strategy.

The results of the study are a contribution to the reality of a developing country like Chile in terms of sustainable food. Future research should include, in addition to local food and nutrition professionals, the opinions of consumers and the community at large. This would provide a more holistic vision for the appropriation of the importance of following a more sustainable diet in the territories and communities of the country.

**Author Contributions:** C.T.-P. and P.C.-R. conceptualization and methodology. C.T.-P. and A.A.-P.: formal analysis, investigation and data curation. C.T.-P.: writing—original draft preparation; P.C.-R., A.A.-P., C.L.-Q. and R.V.: review and editing. P.C.-R. project administration and funding acquisition. All authors have read and agreed to the published version of the manuscript.

**Funding:** This research was funded by Sociedad Chilena de Nutrición–Tetra Pak 2021.

**Institutional Review Board Statement:** The study was conducted in accordance with the Declaration of Helsinki, and approved by the Ethics Committee of Facultad de Medicina, Universidad de Chile, Chile, for studies involving humans.

**Informed Consent Statement:** Informed consent was obtained from all subjects involved in the study.

**Data Availability Statement:** Data available on request due to ethical restrictions. The data presented in this study are available upon request from the corresponding author. The data are not publicly available because this action was not stated in detail to the study participants when signing the informed consent.

**Acknowledgments:** We thank each and every study participant for their valuable contribution to meeting our goals.

**Conflicts of Interest:** The authors declare no conflict of interest.

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
