# Peer review of "Exploring the Meanings of Food Sustainability: An Interpretive Phenomenological Analysis"

_sustainability, doi:10.3390/su151813548_

Round 1
Reviewer 1 Report
The manuscript “Exploring the meanings of food sustainability: An interpretive phenomenological analysis” authored by Claudia et al. is a phenomenological analysis of food security. The manuscript is structured and presented in a good manner. There are the following suggestions to improve the quality of the manuscript-
Modify the abstract for more result centric
Briefly add the status of global food security in the introduction section
There is a clear-cut difference between sustainable food and a sustainable diet so, use carefully both the terminology
Rewrite the sentence “This last situation .. food sustainability with local identities.” for better clarity, line no.67-78
Figure 1 is ok
Revise figure 2 for better clarity
The materials and methods section is well described
The result section is written nicely the interpretation of the results in the discussion section is ok
Figure 3 is ok
Modify the conclusion section for better soundness.
Overall, the manuscript is presented in a nice manner, the present study will also open a new avenue to managing sustainable food and diet for the growing global population. The manuscript may be accepted for publication after suggested modifications/corrections.
Minor English editing required
Author Response
- Reviewer 1
Dear reviewer, thank you very much by your comments. We really appreciate your help.
- Modify the abstract for more result centric
Answer: Abstract is modified in coherence with what was delivered in the manuscript.
- Briefly add the status of global food security in the introduction section
Answer: Thanks for the comment; aspects of food safety are mentioned in the introduction, within the theoretical framework.
- There is a clear-cut difference between sustainable food and a sustainable diet so, use carefully both the terminology
Answer: We appreciate the suggestion. We have modified in the manuscript, appearances that confused the objective of the message.
- Rewrite the sentence “This last situation .. food sustainability with local identities.” for better clarity, line no.67-78
Answer: Wording has been changed for better understanding.
- Revise figure 2 for better clarity
Answer: We appreciate the comment. We have restated the figure, including only the questions that were part of the interview.
- Modify the conclusion section for better soundness.
Answer: Reflection on the conclusions of the manuscript is improved and enriched.
Reviewer 2 Report
Keywords
I suggest including the keyword “food system” instead of one of the included ones.
2. Materials and Methods
This research was carried out using a qualitative investigation approach from an interpretative paradigm of exploratory scope. In my opinion, the authors should explain in more detail this methodology used and that allowed them to make the interpretations and the results obtained.
In addition, I suggest that the authors include bibliographical references on this methodological approach.
On the other hand, I suggest that the authors include bibliographical references on the data analysis.
3. Reults
On lines 163-167 the authors should clarify that the sustainability of a kitchen is determined by a series of sustainability indicators as a whole and not by a single isolated indicator.
4. Discussion
In the Discussion section, I suggest that the authors include further arguments on other social and cultural aspects related to sustainable food or sustainable culinary preparations (for example, artisanal foods, produced foods by poor indigenous families, communities, localities or regions, autochthonous ingredients or traditional recipes).
Limitations
The authors recognize that the study collected valuable information from professionals in the area of sustainable dieting, however, it did not consider the opinion of consumers or the community in general. However, I think that another limitation of the study that the authors should consider is that food sustainability is only one component of the overall food systems sustainability assessment.
Author Response
- Reviewer 2
Dear reviewer, thank you very much by your comments. We really appreciate your help.
- I suggest including the keyword “food system” instead of one of the included ones.
Answer: The keywords are modified; and includes "food systems" for "collective feeding"
- Materials and Methods
This research was carried out using a qualitative investigation approach from an interpretative paradigm of exploratory scope. In my opinion, the authors should explain in more detail this methodology used and that allowed them to make the interpretations and the results obtained.
In addition, I suggest that the authors include bibliographical references on this methodological approach.
On the other hand, I suggest that the authors include bibliographical references on the data analysis.
Answer: We appreciate the suggestion that allows the reader to clarify the methodological aspects. We have detailed in a better way the methodological aspects, including theoretical references that support them.
- Results: On lines 163-167 the authors should clarify that the sustainability of a kitchen is determined by a series of sustainability indicators as a whole and not by a single isolated indicator.
Answer: We appreciate the suggested reflection; we have applicated it in the text of the manuscript.
- Discussion: In the Discussion section, I suggest that the authors include further arguments on other social and cultural aspects related to sustainable food or sustainable culinary preparations (for example, artisanal foods, produced foods by poor indigenous families, communities, localities or regions, autochthonous ingredients or traditional recipes).
Answer: Paragraph and reflection of what was requested is incorporated in the discussion, final part.
- Limitations; The authors recognize that the study collected valuable information from professionals in the area of sustainable dieting, however, it did not consider the opinion of consumers or the community in general. However, I think that another limitation of the study that the authors should consider is that food sustainability is only one component of the overall food systems sustainability assessment.
Answer: We appreciate the suggestion which we have included in the limitations of the study.
Reviewer 3 Report
The paper’s focus needs better streamlining, from talking about sustainable diets to culinary preparations, what exactly is the focus? Make this clearer especially as you say’ the objective of the study was to interpret the meanings given by professionals in the area of food regarding food sustainability’.
How do you then pivot to finding/saying this “The results indicate that, in the sustainability of food services or collective catering, care for the environment is valued in accordance with local regulations in force, although their implementation is difficult due to economic aspects and political will in developing countries”
SUSFANS program -full meaning of acronym
Line 85- ‘Non-probabilistic convenience sampling was used according to saturation point.’ Not sure what you mean here. Do you mean until saturation point was reached?
Your diagram says 66,7 female, who made it up to 100 percent. You use , instead of . but you are writing in English.
What did the participants perceive sustainable diets to mean,
What do you mean by ‘food sustainability with local identities’?.
Given the difficulty in the measurement of sustainability indicators as it concerns diets, how did the participants from the different countries perceive this to mean? What did you as authors define this to entail?
Line 113-115 ‘According to the interviewees, the environmental aspect stands out as an essential part of sustainability in collective food and catering services. It is an element that must be understood from a paradigm that integrates crucial factors for society, economy and also the environment, since it has repercussions on public policies that impact the ecosystem’.
What are these crucial factors for society? Clarify some of the repercussion on public policies.
Line 152-153- It should primarily consist of produce and consider the supply chain sourcing as well as the supply chain itself to be local, seasonal and healthy’- How does this relate to culinary preparation?
I have only pointed out a few of the vague claims which are interspersed throughout the paper. Endeavour to ascertain what it is your paper is focused on and tailor your writing to address the research question or gap your work sets out to fill.
grammar checks and editing needed
Author Response
- Reviewer 3
Dear reviewer, thank you very much by your comments. We really appreciate your help.
- The paper’s focus needs better streamlining, from talking about sustainable diets to culinary preparations, what exactly is the focus? Make this clearer especially as you say’ the objective of the study was to interpret the meanings given by professionals in the area of food regarding food sustainability’.
Answer: We appreciate your opinion and suggestion to our object of study. We have stated in more detail in the objective what was measured in this investigation, in coherence with the results of the latter.
- How do you then pivot to finding/saying this “The results indicate that, in the sustainability of food services or collective catering, care for the environment is valued in accordance with local regulations in force, although their implementation is difficult due to economic aspects and political will in developing countries”
SUSFANS program -full meaning of acronym
Answer: The wording is improved for a better communication of the household items and the definition of SUSFANS is completed
- Line 85- ‘Non-probabilistic convenience sampling was used according to saturation point.’ Not sure what you mean here. Do you mean until saturation point was reached?
Answer: We have improved and further clarified this information in the material and methods section.
- Your diagram says 66,7 female, who made it up to 100 percent. You use , instead of . but you are writing in English.
Answer: We appreciate the suggestion; we have rectified the figure with missing symbol.
- What did the participants perceive sustainable diets to mean
Answer: The perception of participants on culinary preparations and sustainable diets at the beginning of the results is included.
- What do you mean by ‘food sustainability with local identities’?
Answer: Local identity refers to the food or diet typical of a community. It is modified in the text.
- Given the difficulty in the measurement of sustainability indicators as it concerns diets, how did the participants from the different countries perceive this to mean? What did you as authors define this to entail?
Answer: The results show the lack of discrepancy according to the nationality of the study subjects.
- Line 113-115 ‘According to the interviewees, the environmental aspect stands out as an essential part of sustainability in collective food and catering services. It is an element that must be understood from a paradigm that integrates crucial factors for society, economy and also the environment, since it has repercussions on public policies that impact the ecosystem’.
- What are these crucial factors for society? Clarify some of the repercussion on public policies.
Answer: Communication of the role of food and sustainable policies is improved.
- Line 152-153- It should primarily consist of produce and consider the supply chain sourcing as well as the supply chain itself to be local, seasonal and healthy’- How does this relate to culinary preparation?
Answer: The quote is appreciated. The wording in the manuscript is improved.
- I have only pointed out a few of the vague claims which are interspersed throughout the paper. Endeavour to ascertain what it is your paper is focused on and tailor your writing to address the research question or gap your work sets out to fill.
Answer: We appreciate your willingness to improve our manuscript that responds to the research carried out. We hope that the modifications made, manage to respond to your request.
Round 2
Reviewer 2 Report
No comment
Author Response
Thank you for your suggestions. We report in the abstract and in the introduction, more clearly and explicitly, the research question.